# Emerging Drug Targets for Endometriosis

**DOI:** 10.3390/biom12111654

**Published:** 2022-11-08

**Authors:** Marie-Madeleine Dolmans, Jacques Donnez

**Affiliations:** 1Gynecology Department, Cliniques Universitaires St-Luc, Avenue Hippocrate 10, 1200 Brussels, Belgium; 2Gynecology Research Laboratory, Institut de Recherche Expérimentale et Clinique, Université Catholique de Louvain, Avenue Mounier 52, bte B1.52.02, 1200 Brussels, Belgium; 3Department of Gynaecology, Université Catholique de Louvain, 1200 Brussels, Belgium; 4Société de Recherche pour l’Infertilité (SRI), 143 Avenue Grandchamp, 1150 Brussels, Belgium

**Keywords:** endometriosis, GnRH antagonist, inflammation, estrogens, oxidative stress, progesterone resistance, reactive oxygen species (ROS), cytokines

## Abstract

Endometriosis is a chronic inflammatory disease causing distressing symptoms and requiring a life-long management strategy. The objective of this review is to evaluate endometriosis-related pathways and identify novel therapies to treat it. We focused on the crucial role of inflammation and inflammatory molecules in order to define new perspectives for non-hormonal treatment of the disease by targeting inflammation, nuclear factor kappa B and cytokines, or reactive oxygen species, apoptotic and autophagic pathways, regulators of epithelial-mesenchymal transition, and angiogenesis and neuroangiogenesis. Novel non-steroidal therapies targeting these pathways for endometriosis were explored, but multiple challenges remain. While numerous agents have been investigated in preclinical trials, few have reached the clinical testing stage because of use of inappropriate animal models, with no proper study design or reporting of preclinical strategies. Targeting estrogens is still the best way to control endometriosis progression and inflammation.

## 1. Introduction

Endometriosis is a chronic inflammatory disorder causing distressing symptoms like pain and infertility in 5–10% of women of reproductive age [1,2,3]. Each of its three distinct pelvic forms (peritoneal, ovarian and rectovaginal) has specific presentations [2], but dysmenorrhea and chronic non-menstrual pelvic pain are the most common manifestations [3]. Blocking menses by means of hormone therapy to induce oligo- or amenorrhea may in theory modulate the symptoms of endometriosis [1,4]. First-line medical therapies (oral contraceptives [OCPs] and progestogens) do work in two-thirds of women experiencing endometriosis-related pain [5,6,7] while second-line treatments like gonadotropin-releasing hormone (GnRH) agonists and antagonists, which are offered only if OCPs or progestogens fail, are associated with menopausal symptoms [8,9,10].

### 1.1. Concept of Progesterone Resistance

In endometriotic lesions, a lack of secretory transformation during the luteal phase and varying patterns of estrogen receptors (ERs) and progesterone receptors (PRs) in eutopic and ectopic endometrium strongly indicate that PRs, although present, are inactive in a biological sense. Indeed, the notion of progesterone resistance was first proposed back in 1997 [2]. Since then, numerous papers have advanced theories substantiating this hypothesis (see [11] for review). According to Bulun et al [12,13] and Yilmaz and Bulun [14], the fact that endometriotic stromal cells are unable to produce progesterone-induced paracrine factors may be down to a lack of PR-B. In endometriotic implants, ERα is diminished but ERβ activity is upregulated, resulting in complete loss of PR-B, which cannot then induce 17-beta hydroxysteroid dehydrogenase 2 (17β-HSD2). Progesterone receptor status predicts the response to progestogen therapy [15], as also discussed in the review by Reis et al [16].

The origins of progesterone resistance in adult females were reviewed by Donnez and Dolmans [11], with inflammation and oxidative stress found to occupy a central role in both the pathogenesis of endometriosis and progesterone resistance. Erythrocytes, apoptotic endometrial tissue and cell debris inside the peritoneal cavity can potentially induce oxidative stress, while iron overload, oxidative stress and inflammation could all result from lysis of erythrocytes [17,18,19]. Cellular iron storage within ferritin in macrophages may reduce its toxicity, but continuous delivery of iron to macrophages might overload the capacity of ferritin to store iron, triggering oxidative stress [18,19]. Proinflammatory factors, cytokines and interleukin 1 beta (IL-1β) may disrupt PR function and contribute to progesterone resistance, ultimately resulting in increased nuclear factor kappa B (NFκB) involvement in endometriosis development [15,16,20].

A genetic basis and epigenetic influence [21,22] along with neonatal progesterone resistance may of course affect the degree of progesterone resistance, as discussed in several reviews [11,16,23,24,25,26]. Indeed progesterone resistance could be due to a range of factors, including DNA hypermethylation [11].

### 1.2. Heterogeneity of Lesions

Among peritoneal lesions, active red lesions are known to be deeply infiltrated by macrophages residing in the stroma [27]. They are resistant to progestogen therapy, possibly provoking some degree of decidualization, but not atrophy [2,28,29]. Numerous differences have been identified between eutopic and ectopic endometrium, which may be explained by different patterns of steroid receptors [2,12,13,14]. The heterogeneity of lesions and their intervariability during the luteal phase were first reported by Nisolle and Donnez [2] and reviewed by Redwine [30] in 2002. As reexamined in recent papers, the histological characteristics of superficial peritoneal lesions are rarely in phase with those observed in eutopic endometrium [31,32]. This heterogeneity and intervariability, known for more than 25 years now [2,30], is probably linked to progesterone resistance in endometriotic lesions, as recently reviewed [11,25].

### 1.3. Crucial Role of Inflammation and Inflammatory Molecules

Inflammation is one of the key mechanisms triggering any disease involving invasion and dissemination [17,20], so endometriosis, which induces cell proliferation and infiltration, is no exception in that sense. Macrophages play a major role in this process by presenting antigens to T cells and initiating inflammation by releasing cytokines that activate other cells (27]. In the context of endometriosis, their role is pivotal in lesion adhesion, proliferation, vascularization, angiogenesis, neuroangiogenesis and sensitization of nerves [20].

Numerous growth factors, including fibroblast growth factor, macrophage-derived insulin-like growth factor, platelet-activating factor and vascular endothelial growth factor (VEGF), are more abundant in ectopic sites of endometriosis patients than healthy subjects [33,34,35]. Migration inhibitory factor (MIF) and monocyte chemoattractant protein-1 are two of the most potent factors behind endometriosis-related inflammation. They recruit macrophages into endometriotic lesions, aiding their further growth and proliferation by releasing proinflammatory cytokines and a variety of other growth factors [36,37].

## 2. From Pathophysiology to New Perspectives (Figure 1)

As our understanding of the pathophysiology of endometriosis advances, so do our perspectives for novel non-hormonal treatments for the disease, with non-steroidal anti-inflammatory drugs and antiangiogenic agents considered frontrunners. However, there is a long way to go from preclinical trials to clinical testing. According to Vanhie et al [38], while numerous agents have undergone testing in preclinical trials, relatively few have reached the clinical testing stage due to use of inappropriate animal models and lack of proper study design and reporting of preclinical strategies.

**Figure 1 biomolecules-12-01654-f001:**
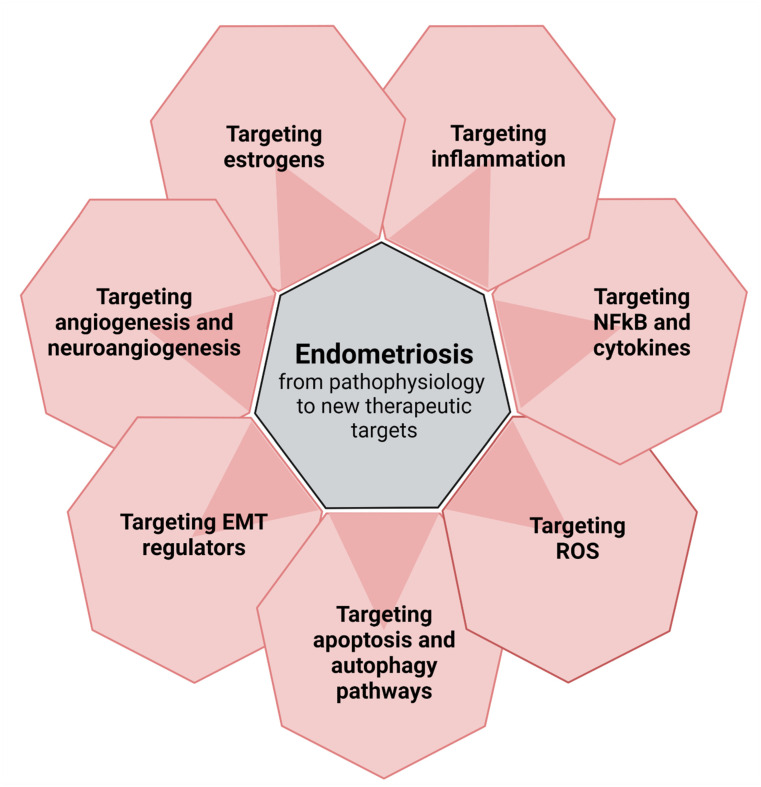
From pathophysiology to new perspectives: targets for new therapies.

### 2.1. Targeting Inflammation

#### 2.1.1. Prostaglandin E2, Cyclooxygenase-2 and Tumor Necrosis Factor-α

In a recent paper, Yu et al discussed the role of prostaglandin E2 (PGE2) in the development of endometriosis [39]. PGE2 is an eicosanoid produced by cyclooxygenase (COX)-2, and increased expression of COX-2 in endometriotic lesions plays a role in the evolution of the disease [40], while IL-1β regulates COX-2 expression in endometriosis. The COX-2 gene is more sensitive to IL-1β stimulation in ectopic endometriotic stromal cells than in eutopic stromal cells [40] and the COX-2/PGE2 pathway is closely related to endometriosis [41]. COX-2 expression is associated with multiple transcriptional pathways and involvement in various pathological processes that include inflammation, cancer and multidrug resistance [41,42]. COX2/PGE2 signaling may also be directly implicated in the pathogenesis of endometriosis, including regulation of ectopic implantation and growth of the endometrium, angiogenesis and immunosuppression [40,41]. Expression of COX-2 is rapidly upregulated in response to diverse proinflammatory signals and it plays a significant role in the origin and progression of endometriosis [42]. In women with the disease, expression of COX-2 was higher in endometrial glandular epithelium, endometrial stroma and peritoneal fluid than in women without endometriosis [43].

PGE2 is a regulator of the immune response and exerts two opposing effects: inflammatory or anti-inflammatory. In endometriosis, inflammatory mediators (COX-2/PGE2) are the target of non-steroidal anti-inflammatory drugs (NSAIDs) [39]. However, anti-inflammatory drugs interfere with the function of COX-1 and -2 enzymes, inhibiting production of prostaglandins. Selective COX-2 inhibitors (celecoxib, rofecoxib and valdecoxib) have been investigated [20,38], with celecoxib found to reduce the number, volume and vascularization of endometriotic lesions in mice [44]. Unfortunately, as reviewed by Kapoor et al [20], some COX-2 inhibitors (rofecoxib, valdecoxib) had to be withdrawn from the market due to serious side effects, including myocardial infarction and stroke. 

Tumor necrosis factor alpha (TNF-α) is a proinflammatory cytokine that plays a role in endometriosis by stimulating adhesion of endometrial cells and their ectopic proliferation [42]. The anti-inflammatory effect of blocking TNF-α by monoclonal antibodies (e.g. infliximab) or soluble TNF-α receptors (e.g. etanercept) has been demonstrated in vivo in animal models and also in humans.

D’Hooghe et al [45] tested the hypothesis that recombinant human TNFRSF1A (r-hTBP1) can inhibit development of endometriotic lesions in baboons, an established model for endometriosis studies. In animals with laparoscopically confirmed endometriosis, blockading TNF-α with p55-soluble TNF-α receptors results in inhibition of progression and growth of endometriotic implants [45]. The surface area of endometriotic lesions was found to be smaller and the disease less severe in r-hTBP1-treated baboons [45].

In rats with ectopically transplanted endometrial tissue, administration of r-hTBP-1 resulted in defective development of implants compared to controls [46]. Nevertheless, in a randomized placebo-controlled trial, Koninckx et al [47] failed to demonstrate that infliximab, an anti-TNF- α monoclonal antibody, relieves pain.

#### 2.1.2. Targeting NFκB and Cytokines

Natural killer cells are one of the most important entities in the innate immune system. Their diminished activity in endometriotic lesions is to blame for their decreased efficiency in clearing endometriotic cells from the peritoneal cavity of women [17,20]. NFκB is a transcription factor regulating innate immunity and also controlling transcription of DNA cytokine production and cell survival at the cellular level. 

In endometriotic cells, NKκB signaling is activated by stimuli like TNF-α and IL-1β [48]. The transcriptional activity of a number of proinflammatory cytokines/chemokines, such as IL-1, IL-6, IL-8, TNF-α, RANTES, MIF and ICAM1, is activated by NFκB signaling, demonstrating the key role of NFκB in the inflammatory response in endometriosis [49,50]. NFκB signaling is related to progression of endometriosis through several key factors, including estrogen, progesterone, oxidative stress and noncoding miRNAs (nc miRNAs), and can regulate the cellular behavior of endometriotic cells and peritoneal macrophages in the endometriotic milieu. NFκB-activated macrophages release proinflammatory cytokines and growth factors implicated in boosting levels of inducible nitric oxide synthase, COX-2, IL-1, IL-6, IL-8, TNF-α and VEGF, which is why pharmacological inhibitors targeting NFκB could represent a possible therapeutic approach. Liu et al [48] reviewed potential anti-NFκB drugs for endometriosis treatment.

There is no doubt that endometriosis patients show elevated levels of expression and release of a range of proinflammatory cytokines and growth factors, including IL-1, IL-6, IL-8, epidermal growth factor and hepatocyte growth factor, in their ectopic and eutopic endometrium and peritoneal fluid [20]. Among various pharmacological substances antagonizing the impact of these cytokines, tocilizumab may prove effective at reducing endometriosis-associated inflammation [51]. Kapoor et al [20] recently published a record of pharmacological inhibitors that target inflammatory molecules, making them key players in mitigating endometriosis. Resveratrol downregulates in vitro expression of inflammatory markers in eutopic endometrium, but even more so in ectopic endometrium [20]. Tocilizumab [51], a monoclonal anti-IL-6 antibody, causes lesions to regress in rats. Pyrvinium pamoate [52] targets IL-6 and IL-8 and suppresses their mRNA expression in vitro. Nobiletin [53], acting upon NFκB, IL-6 and IL-1β, decreases lesion size and thereby levels of pain by inhibiting cell proliferation, angiogenesis and excess inflammation in mice. All these drugs have only been used in rodent models or with in vitro endometrial cell culture.

#### 2.1.3. Chronic Inflammation and Epigenetics

There is increasing evidence that epigenetics and particularly regulatory DNA methylation mechanisms play a role in the pathophysiology of endometriosis [54]. In the presence of chronic inflammation, immune cells may be able to trigger changes in epigenetic regulation, resulting in hypermethylation. Among transcription factors responsible for uterine function, HOXA10 is widely known to be associated with endometrial receptivity and progesterone receptor expression, and is hypermethylated in patients with endometriosis [55]. A recent study demonstrated the effectiveness of 5’-aza-deoxycytidine (AZA), a DNA methylation inhibitor, on HOXA10 methylation levels in vitro. All this provides important proof of concept and forms the basis for development of future therapeutic strategies [56]. Unfortunately, existing DNA methylation inhibitors, recently approved by the FDA for hematological malignancies, are far from ready for clinical application in patients with endometriosis. Indeed, these drugs may be toxic to hematopoietic and gastrointestinal systems, and are therefore contraindicated in patients wishing to conceive.

### 2.2. Targeting Reactive Oxygen Species

Increased reactive oxygen species (ROS) levels associated with chronic inflammation in endometriosis are also due to ROS detoxification pathway dysregulation [17,18,19,20]. ROS may therefore function as signaling molecules to maintain an endometriosis-related proliferative phenotype [17]. They actually exert influence as second messengers for cell proliferation by triggering growth-related signaling pathways like the serine/threonine protein kinase/mitogen-activated kinase/extracellular signal-regulated kinase (RAF/MEK/ERK) network, which participates in proliferation in response to higher endogenous ROS levels [57,58]. ERK 1 and 2, part of the MAPK family, are located in the cytoplasm and undergo nuclear translocation when activated by MEK 1 and 2 to promote gene expression for cellular proliferation, survival, differentiation and adhesion. Downregulation of ERK 1 and 2 activation by first-generation MEK 1 and 2 inhibitors caused a decline in both human endometrial cell proliferation in vitro and human endometriotic lesions grafted to nude mice [59]. Other agents suppressing the same pathway were also investigated in murine models of endometriosis [15,20].

Leflunomide, serving as a tyrosine kinase inhibitor to curb NFκB transcription, and sorafenib, a multikinase inhibitor targeting serine/threonine RAF kinases (RAF-1 and B-RAF), were found to curtail proliferative activity in ectopic endometrial cells [59,60,61]. However, sorafenib administration yielded contradictory results in terms of antiproliferative and antiangiogenic properties in rat models of endometriosis [62,63], which may have been due to bias from the animal models themselves. Inconsistencies in stromal and endometrial cell regulators in endometriotic lesions could also explain this discrepancy, as ectopic stromal cell proliferation appeared to be less related to ROS-dependent ERK 1 and 2 upregulation, hinting at other pathways keeping cells in a hyperproliferative state [64]. One of them is the phosphoinositol 3-kinase/protein kinase B/mammalian target of rapamycin (PI3K/Akt/mTOR) pathway, which regulates multiple cell functions like growth, survival and metabolism, and plays a pivotal role in ovarian and deep nodular endometriosis [65].

Cannabinoid agonists that inhibit both the RAF/MEK/ERK and PI3K/Akt/mTOR pathways are also capable of suppressing proliferation in case of deep nodular endometriosis [66], clearly demonstrating the involvement of numerous proliferative pathways in lesion growth and development. According to Cacciottola et al [17], some of the cited drugs with antiproliferative and antiangiogenic properties are already used in humans for cancer indications. However, because of the high risk of side effects and their incompatibility with reproductive function and chances of conception, they are considered unsuitable for treatment of endometriosis, particularly in young patients.

Various phytochemicals have also been tested [67,68], including naringenin, an antioxidant with antiproliferative and antioxidant properties that favors apoptosis. Lesion growth was indeed suppressed in endometriosis models through modulation of this pathway [20,69,70]. Other antioxidants can be extracted from plants. Studies in rodents have demonstrated that curcumin can decrease development of endometriotic lesions, but this was not confirmed in experiments with human endometrial cells [71,72].

N-acetylcysteine (NAC) exhibited antioxidant and antiproliferative effects in animal models of endometriosis by downregulating ERK 1 and 2 kinase activity [73,74]. Treatment with NAC significantly reduced endometriotic cyst volume and endometriosis-related pain [75] in women diagnosed with ovarian endometriosis.

### 2.3. Targeting Apoptotic and Autophagic Pathways and Tumor-Promoting Genes/Proteins

The uterus has a coordinated system of ridding itself of unwanted cells, shedding its endometrium under the control of estrogens and progesterone. In normal endometrium, apoptotic proteins increase during the late luteal phase, but in endometriosis patients, they do not, so apoptosis is inhibited and endometrial cells may survive and implant in ectopic sites [20,70].

One aspect hampering expression of apoptotic proteins is an upturn in cytoprotective enzymes, which are activated in the presence of oxidative stress [17,20,70]. Autophagic proteins are also crucial to survival of endometriotic cells and, in particular, endometriosis recurrence. Apoptotic and autophagic pathways may therefore prove to be promising targets for creation of new therapeutic alternatives to treat endometriosis [20,70,76].

Recent research alludes to the possibility of acting against endometriotic cells via autophagy-apoptosis pathways. Indeed, Mao et al suggested that silencing cir-RNA (007299) promotes apoptosis of ectopic endometriotic cell sites [77]. Sapmaz et al [78] demonstrated that some drugs like metformin, letrozole, and atorvastatin induced apoptosis and anti-inflammatory effects in experimental models of both ovarian and peritoneal endometriosis. According to Lin et al [79], increased ERα signaling and decreased PR-B expression synergistically led to a hypoautophagic state in ectopic endometrial stromal cells, which further inhibited their apoptosis. This study showed that the apoptotic effect of SCM-198, a pseudoalkoloid and synthetic form of leonurine, on ectopic endometrial stromal cells was achieved by reversing the ERa/PR imbalance, upregulating autophagic activity and inducing apoptosis.

### 2.4. Targeting Regulators of Epithelial-Mesenchymal Transition

Epithelial-mesenchymal transition (EMT) is a process that occurs in cell remodeling, during which cells are transformed and become invasive, similar to chronic inflammation, fibrosis and cancer progression. EMT is characterized by the gradual loss of the epithelial phenotype and progressive gain of a mesenchymal phenotype. In a recent review by Konrad et al [80], many EMT-specific pathways like Twist, Snail, Slug, Zinc finger E-box-binding homeobox 1/2 (ZEB1/2), E/N-cadherin, keratins and claudins were described. The majority of changes in EMT-related marker expression were detected in ectopic endometrium, especially in the three identified types of endometriosis, namely ovarian, peritoneal, and deep-infiltrating disease, compared with eutopic endometrium. According to Konrad et al [80], only partial EMT occurs in endometriosis, but there is no doubt that EMT is involved in endometriosis development. Endometrial cells are able to change their structural and functional state from a polarized epithelial phenotype to a highly motile mesenchymal phenotype, with E-cadherin and N-cadherin emerging as key players [17]. Two principal factors are responsible for the EMT process in endometrial cells: the hypoxic environment and estrogen concentrations [75,76].

By targeting 76 genes, Suda et al [81] demonstrated different mutation profiles between paired epithelial and stromal cells in both ovarian endometriomas and normal endometrium. They suggested, like Noë et al [82], that the origins of epithelial and stromal cells are independent of one another, at least in ovarian endometriomas, and remain skeptical of full-scale EMT. EMT is often considered a primary mechanism for cell migration and invasion, but collective cell migration (CCM) has also been identified in endometriotic lesions in baboons and is involved in the pathogenesis of deep endometriosis [83,84]. In CCM, cells bind to the extracellular matrix and migrate as a collective rather than individual cells, thereby avoiding EMT.

Since EMT/CCM are involved in endometriosis development, drugs targeting cell migration could be used for endometriosis. A recent review by Liu et al [85] reported that EMT markers are clearly expressed in ectopic endometrium and are closely associated with estrogens. Pharmacological inhibitors targeting EMT regulators could prove beneficial to endometriosis patients. These inhibitors include isoliquiritigenin [86], fucoidan [87], melatonin [88] and 3,6-dihydroxyflavone [89], but have so far only been tested in murine endometrial cell lines. The same is true of drugs targeting CCM, which have only been used in cancer cell lines. Studies have shown that CCM can be inhibited in cancer cells, with ursolic acid curbing CCM in glioblastoma [90], gold nanorods and near-infrared light impeding CCM in breast cancer cell lines [91], and SU6656, an Src inhibitor, hindering invasion by melanoma cells by inhibiting CCM [92]. However, there have been no investigations into endometrial or endometriotic cells in this context to date.

### 2.5. Targeting Angiogenesis and Neuroangiogenesis

The process of forming new blood vessels and ensuring a proper blood supply is termed angiogenesis. Among the most critical factors stimulating angiogenesis is VEGF, which facilitates migration of growing cells, their vascularization and even their invasion. As new blood vessels form, different growth factors and the matrix metalloproteinase (MMP) complex are expressed. Expression of VEGF is increased in red peritoneal lesions and peritoneal fluid from endometriosis patients. [29]. There are various other factors also relevant to angiogenesis, such as platelet-derived endothelial cell growth factor, endoglin, MIF, ILs, and protein tyrosine phosphatase [20]. Some drugs, like sunitinib, SU6668, SU5416, sorafenib and pazopanib, interfere by inhibiting angiogenic pathways to reduce lesion formation [93,94]. Quinagolide was found to substantially decrease lesion size, potentially by regulation of angiogenesis [95].

Dopamine and dopamine receptor 2 agonists promote VEGF receptor endocytosis and are able to reduce neoangiogenesis [96,97]. In a recent review, Pellicer et al [96] showed that targeting angiogenesis with dopamine agonists (DAs) is a promising strategy for endometriosis patients. DAs (bromocriptine, cabergoline and quinagolide) downregulate proangiogenic but upregulate antiangiogenic pathways, thereby blocking cellular proliferation of endometriotic lesions.

Several studies in baboons have demonstrated the crucial role of collective cell migration in the invasion process [98] and suggested that nerve fibers may participate in the development and invasion process of endometriotic lesions [83,84,99]. Neuroangiogenesis (co-recruitment of nerves and blood vessels) is believed to play an integral part in the establishment and growth of endometriotic lesions. Sun et al [100] demonstrated that endometrial stromal cells isolated from eutopic endometrium of women with endometriosis could secrete exosomes, which have an important role in the development of endometriosis by regulating neuroangiogenesis. According to a review by Saunders and Horne [101], the link between growth of new blood vessels and nerve fibers has provided a mechanism that could explain not only the association between the presence of ectopic tissue and pain pathways, but also the cross-talk between nerves and immune cells with neuroinflammation. This strongly suggests that nerves are involved in the pathogenesis of endometriosis.

Finally, according to a recent review by Vannuccini et al [102], estrogen dependency and progesterone resistance are the key events that cause ectopic implantation of endometrial cells, decreasing apoptosis and increasing oxidative stress, inflammation and neuroangiogenesis. Inhibiting neuroangiogenesis in peritoneal and deep endometriotic lesions could prevent proliferation and invasion by ectopic glands. However, as stressed by Vanhie et al [38] and Kapoor et al [20], there are still challenges ahead in the development of novel non-steroidal therapies for endometriosis, while non-hormonal agents have not yet made their way into routine clinical practice.

### 2.6. Targeting Estrogens

Properly balanced concentrations of estrogen and progesterone serve to maintain the regular physiology and functioning of eutopic endometrium during normal menstrual cycles. Estrogen governs endometrial proliferation, while progesterone inhibits the effects of estrogen and helps set in motion the process of decidualization. Any imbalance between progesterone and estrogen results in impaired uterine and endometrial functions [11,103]. PR deficiency is apparent in endometriotic lesions, leading to progesterone resistance and defective progesterone action, and clearly affecting survival and development of endometriotic tissues [11,12,13,14].

Theoretically, the optimal solution would be to reduce estradiol (E2) levels just enough to induce amenorrhea and treat symptoms, while at the same time maintaining adequate concentrations to alleviate vasomotor menopausal symptoms (mainly hot flushes) and bone mineral density (BMD) loss [11,103]. Barbieri’s threshold hypothesis [104] states that partially suppressing E2 to within the 30–60 pg/mL range may offer the best existing compromise between efficacy, tolerance and safety [99]. Knowing that estrogens play a critical role in survival and vascularization of endometriotic implants, it would be wholly appropriate to consider lowering their concentrations as a treatment approach. GnRH antagonists induce competitive blockage of the GnRH receptor, dose-dependently suppressing production of follicle-stimulating hormone (FSH) and luteinizing hormone (LH), and inhibiting secretion of ovarian steroid hormones without triggering a flare-up effect [103,105]. The main advantages of these new drugs are dose-dependent estrogen suppression (from partial at lower doses, to complete at higher doses) and prompt reversibility.

As recently reviewed [103], three oral GnRH antagonists, elagolix, relugolix and linzagolix, have generated robust data in the latest randomized, placebo-controlled clinical trials for treatment of endometriosis-associated pain in symptomatic premenopausal subjects. One of them, elagolix, has been approved by the FDA for the management of endometriosis-related pain [106,107,108].

The mean plasma half-life (t1/2) of elagolix ranges from 2.4 to 6.3 h [106,107,108]., theoretically requiring the drug to be taken twice daily. The efficacy of 6 months of treatment with elagolix was evaluated. Two different regimens of the drug (150 mg once daily and 200 mg twice daily) were tested. Administration of 200 mg elagolix twice daily induced strong suppression of E2 and yielded marked improvements in dysmenorrhea and non-menstrual pelvic pain, albeit at the expense of increased hot flushes and a more pronounced decline in BMD. Ongoing studies are evaluating the impact of add-back therapy (ABT) administered along with elagolix.

Linzagolix has a half-life of 15–18 h [103,109,110]. Three separate doses (75 mg, 100 mg, and 200 mg once daily for 24 weeks) were investigated. Able to achieve full suppression of serum E2 to postmenopausal levels, the once-daily 200 mg linzagolix dose proved very effective at managing endometriosis-related dysmenorrhea and non-menstrual pelvic pain (same efficacy as the 100 mg group), but also had a significant impact on dyspareunia and some aspects of quality of life. In addition, higher rates of hypoestrogenic symptoms were encountered, including BMD loss of ≥3% in some patients after 24 weeks. Clearly, this once-daily dose would require hormone ABT if used for over 6 months [109].

Relugolix has a half-life of 37–42 hours [111,112] and was administered once daily at a dose of 40 mg with ABT (1 mg E2 and 0.5 mg norethisterone acetate [NETA]). This once-daily regimen was found to significantly reduce dysmenorrhea and non-menstrual pelvic pain in endometriosis patients. Thanks to associated ABT, relugolix combination therapy was well tolerated, rates of vasomotor symptoms were similar to a placebo, and BMD was maintained for 24 weeks [112].

## 3. Conclusions

The pathogenesis of endometriosis remains a source of contention. Every breakthrough in our understanding of the pathophysiology of endometriosis increases our prospects of finding the right non-hormonal treatment for the disease, with non-steroidal anti-inflammatory drugs and antiangiogenic agents considered frontrunners. However, there is a long way to go from preclinical trials to clinical testing.

Targeting inflammation, NFκB and cytokines, ROS, apoptotic and autophagic pathways, and tumor-promoting genes/proteins should be the goal of future research. Anti-inflammatory drugs interfere with the function of COX 1 and 2 enzymes, inhibiting production of prostaglandins. Pharmacological inhibitors targeting NFκB may serve as potential therapeutic agents, with recent research focusing on the influence of NFκB-activated macrophages on different mechanisms, and their subsequent impact on endometriosis progression by release of proinflammatory cytokines and growth factors. Some drugs with antiproliferative and antiangiogenic properties have already been authorized for use in cancer patients. However, because of the high risk of side effects and their incompatibility with reproductive function and chances of pregnancy, they are inappropriate for endometriosis treatment, especially in young subjects.

Apoptotic and autophagic pathways may also be legitimate objectives for development of new therapeutic alternatives to treat endometriosis. EMT is a crucial process wherein cells are structurally transformed and become invasive, as seen in case of chronic inflammation and fibrosis, as well as cancer progression. Pharmacological inhibitors that target these EMT regulators could be of considerable benefit to endometriosis patients. One of the key factors stimulating angiogenesis is VEGF, so targeting angiogenesis with dopamine agonists may well prove a promising strategy. However, as stressed in this manuscript, developing novel non-steroidal therapies for endometriosis is still proving a challenge, as non-hormonal agents are not yet approved in routine clinical practice. As progesterone resistance may also be due to hypermethylation [113], the evaluation of demethylation agents to treat endometriosis may be a future research approach [23].

Lowering E2 concentrations enough to manage symptoms, while maintaining sufficient concentrations to alleviate menopausal issues (principally hot flushes) and BMD loss, may well be the ideal solution. Oral GnRH antagonist induces dose-dependent estrogen suppression, allowing modulation of E2 levels and raising the possibility of individual treatment tailoring.

Endometriosis requires sustained life-long management. The aim is to maximize use of medical therapy and avoid multiple surgical procedures. Healthcare professionals involved in pharmaceutical research and clinical trials must acquire and be able to provide a complete overview of the impacts and side effects of existing therapies. Finally, all areas that require further scrutiny in terms of treatment efficacy and safety need to be thoroughly investigated in real-world populations.

## Data Availability

Not applicable.

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
