# Peer review of "Emerging Drug Targets for Endometriosis"

_biomolecules, 2022, doi:10.3390/biom12111654_

Round 1

Reviewer 1 Report (Previous Reviewer 3)

General remarks

As stated in the authors replies 'the authors are not sure that this reviewer has read carefully our paper' is a guess which in my opinion is arrogant and should be avoided.

Major issues

Lines 239-240 – It remains very unclear how such general mechanisms like apoptosis and/or autophagy which are in almost all organs might become promising new therapeutic alternatives. I miss suggestions from the authors. How do the authors think it might be possible to treat apoptosis only in the endometriotic lesions without severe side effects – maybe locally (see below)?

Chapter 2.5. – Although I criticized this paragraph and suggested several things, the authors ignored Suda et al. 2019 Hum Reprod 34:1899 who stated that a complete EMT from epithelial to stromal cells (also not vice versa) was not observed in endometriosis.  I want to add Noë et al. 2018 I Pathol 245:265 who also have been very skeptical about a full-scale EMT. Up to date a full cellular transition like in cancer cells was never shown in endometriosis. The authors should be more critical in this paragraph and also in the conclusion (lines 355-357).

Chapter 2.6. lines – 280-282 '…and have suggested that nerve fibers may participate in the development and invasion process of endometriotic lesions (84-87)'. In the ref. no. 85 (Fertil Steril 2015;104:491) I could not find any description of nerve fiber density; obviously I have not only read the manuscript carefully but also the self-citation(s) of the authors.  In ref. no. 87 (Fertil Steril 2013;100:1144) the authors found a lower nerve fiber density in deep infiltrating nodules after 6 months but after one year as shown in ref. no. 84 (Fertil Steril 2017;107:987) the nerve fiber density in the deep nodules was higher.  It remains unclear how nerve fiber density should contribute to the invasion process of endometriotic lesions. Please give some arguments. Furthermore, the authors have never shown any causal relationships of nerve fiber density with the development and the invasion process  of endometriotic lesions. The authors should also add a remark that most of their data have been obtained with baboon models, because in the opinion of the authors  '… animal models…fail to reflect the true pathophysiology of endometriosis.'

Discussion/Conclusion – Although I suggested the authors to think about how to overcome PR resistance, however, the authors wondered why 'this reviewer suggest to trigger expression of PR as they are probably biologically inactive'. As published by several groups PR resistance might be also due to DNA hypermethylation (e.g. RBMO 2021;43:139; Reprod Sci 2019;26:1568; Epigenetics 2006;1:106), therefore one might think about treatments to overcome hypermethylation or to prevent it by inhibitors. Epigenetic changes are reversible. Additionally, also Hand2, a mediator of PR action can be impaired in endometriosis and it was shown that expression of Hand2 can be upregulated by FGF inhibitors (Reprod Sci 2019;26:979).

I also wonder why the authors do not understand '…what this referee means by local therapies'. It is well known that besides pain generated by the endometriotic lesions very often the uterus might also generate pain. Thus, one might think also about local therapies like intrauterine applications [e.g. local application of danazol (Fertil Steril 2006;85:Suppl.1:1157) or other agents] or maybe via the peritoneal fluid. 

Minor issues

Line 12 - Please replace pathway by pathways and additionally the end of this sentence is confusing.

Line 33 – and second-line treatments like… sounds better

Line 74 – Ref 32 is not a review

Line 116 – Ref 43 This review deals mainly with Cox-2 inhibitors, maybe another review might be better suited

Line 143 – failed instead of fail

Line 254 – the ref.no.17 seems to be wrong

References – Several numbers are missing (e.g. 97) and others are still with et al. after 3 authors (e.g. 45)

Author Response

Answer to comments of Reviewer 1:

General remarks

As stated in the authors replies 'the authors are not sure that this reviewer has read carefully our paper' is a guess which in my opinion is arrogant and should be avoided.

This reviewer finds that one of our reply is arrogant ... we did not mean to be arrogant, but only judged some of the reviewer's criticisms and comments not appropriate.

Please see below our answers.

Major issues

Lines 239-240 – It remains very unclear how such general mechanisms like apoptosis and/or autophagy which are in almost all organs might become promising new therapeutic alternatives. I miss suggestions from the authors. How do the authors think it might be possible to treat apoptosis only in the endometriotic lesions without severe side effects – maybe locally (see below)?

We understand the comment but we do not fully agree with it as several papers published in 2022 by Mao et al, Sapmaz et al and Lin et al (and there are numerous others, see Pubmed) clearly described new therapeutic alternatives acting on apoptosis and autophagy in endometriosis. The authors have modified the ms by adding some sentences on research evoking the possibility to act against endometriotic cells via the autophagy-apoptosis pathways and their related references.

Mao et al suggested that silencing of cir-RNA (007299) promotes apoptosis of ectopic endometriotic cells.

Sapmaz et al demonstrated that some drugs like metformin, letrozole, and atorvastatin showed apoptosis induction and anti-inflammatory effects on both ovarian and peritoneal endometriosis in experimental models.

According to Lin et al, increased estrogen-estrogen receptor-α signaling and decreased progesterone receptor isoform B (PRB) expression synergistically led to a hypo-autophagy state in eESCs, which further inhibited the apoptosis of eESCs. Their study demonstrated that the apoptotic effect of SCM-198 on eESCs was attained by upregulating the autophagy level.

Chapter 2.5. – Although I criticized this paragraph and suggested several things, the authors ignored Suda et al. 2019 Hum Reprod 34:1899 who stated that a complete EMT from epithelial to stromal cells (also not vice versa) was not observed in endometriosis.  I want to add Noë et al. 2018 I Pathol 245:265 who also have been very skeptical about a full-scale EMT. Up to date a full cellular transition like in cancer cells was never shown in endometriosis. The authors should be more critical in this paragraph and also in the conclusion (lines 355-357).

We agree with this comment. We have added Suda et al and Noe et al.... but we would like to mention that Suda et collected only 11 ovarian endometriotic samples and it is well known that ovarian endometriosis is a different entity which shows different pathological findings (see the "old" paper by Czernobilsky et al) which are not present in peritoneal or deep endometriosis. It may explain some of their conclusions :"targeting 76 genes, it was suggested that epithelium and  stromal cells did not share mutations ....."

Same remark and conclusions from the study of Noê et al.

We may understand that the reviewer is skeptical about a full-scale EMT.... nevertheless, we would like to add that several recent papers (see the review by Liu et al, 2022) described that EMT markers are clearly expressed in ectopic endometrium and are closely associated with estrogen.

In conclusion, we have modified the ms and stressed that there are arguments pros and con EMT and added the related references.

Chapter 2.6. lines – 280-282 '…and have suggested that nerve fibers may participate in the development and invasion process of endometriotic lesions (84-87)'. In the ref. no. 85 (Fertil Steril 2015;104:491) I could not find any description of nerve fiber density; obviously I have not only read the manuscript carefully but also the self-citation(s) of the authors.  In ref. no. 87 (Fertil Steril 2013;100:1144) the authors found a lower nerve fiber density in deep infiltrating nodules after 6 months but after one year as shown in ref. no. 84 (Fertil Steril 2017;107:987) the nerve fiber density in the deep nodules was higher.  It remains unclear how nerve fiber density should contribute to the invasion process of endometriotic lesions. Please give some arguments. 

In theory, we agree with the comment and it is difficult to prove (today) that nerves contribute to the invasion process..... but they are at least responsible for pain (Anaf et al).

Moreover, in a manuscript describing new targets, we consider logical and appropriate to extrapolate some hypothesis.

We have added some arguments about neuroangiogenesis from the recent literature.

“Neuroangiogenesis (co-recruitment of nerves and blood vessels) is believed to play an integral part in the establishment and growth of endometriotic lesions and Sun et al demonstrated that endometrial stromal cells isolated from eutopic endometrium of women with endometriosis could secrete exosomes which play an important role in the development of endometriosis by regulating neuroangiogenesis (Sun et al).”

“According to a review by Saunders and Horne, the link between the growth of new blood vessels and nerve fibers (namely neuroangiogenesis) has provided a mechanism that could explain not only the association between the presence of ectopic tissue and pain pathways but also the cross-talk between nerves and immune cells with neuroinflammation, strongly suggesting the implication of nerves in the pathogenesis of endometriosis.“ 

“According to a recent review by Vannuccini et al, estrogen dependency and progesterone resistance are the key events which cause the ectopic implantation of endometrial cells, decreasing apoptosis and increasing oxidative stress, inflammation and neuroangiogenesis.”

Furthermore, the authors have never shown any causal relationships of nerve fiber density with the development and the invasion process of endometriotic lesions. The authors should also add a remark that most of their data have been obtained with baboon models

Not only ....as the baboon model conclusions were confirmed in human deep endometriotic lesions. Garcia et al who documented that invasion was associated with collective cell migration and nerve development, because in the opinion of the authors  '…most animal models…fail to reflect the true pathophysiology of endometriosis. as in most models, endometriosis is induced by auto transplantation of superficial endometrium.(see Dehoux , Fertil Steril)

We added some sentences to be more critical, as suggested by the reviewer.

Discussion/Conclusion – Although I suggested the authors to think about how to overcome PR resistance, however, the authors wondered why 'this reviewer suggest to trigger expression of PR as they are probably biologically inactive'. As published by several groups PR resistance might be also due to DNA hypermethylation (e.g. RBMO 2021;43:139; Reprod Sci 2019;26:1568; Epigenetics 2006;1:106), therefore one might think about treatments to overcome hypermethylation or to prevent it by inhibitors

As we have written an extensive review on PR resistance in JCM 2021, we did not want to repeat some sentences and be accused of plagiarism. Of course, in our review, we mention that PR resistance may be due to a lot of factors including DNA hypermethylation.

The goal of the present paper was not to "re"explain in detail the causes of PR resistance...
According to the reviewer’s remark, we added sentences on DNA hypermethylation and epigenetics changes.

Epigenetic changes are reversible. Additionally, also Hand2, a mediator of PR action can be impaired in endometriosis and it was shown that expression of Hand2 can be upregulated by FGF inhibitors (Reprod Sci 2019;26:979). Added

I also wonder why the authors do not understand '…what this referee means by local therapies'. It is well known that besides pain generated by the endometriotic lesions very often the uterus might also generate pain. Thus, one might think also about local therapies like intrauterine applications [e.g. local application of danazol (Fertil Steril 2006;85:Suppl.1:1157) or other agents] or maybe via the peritoneal fluid. 

We added some sentences about local therapies although the results (including those related to local danazol) are either very controversial or not confirmed.

Minor issues

Line 12 - Please replace pathway by pathways and additionally the end of this sentence is confusing. corrected

Line 33 – and second-line treatments like… sounds better. We agree. Modified accordingly

Line 74 – Ref 32 is not a review. I agree. Ref 32 is a letter. Ref 32 was put on another line to avoid confusion with the term "review". 

Line 116 – Ref 43 This review deals mainly with Cox-2 inhibitors, maybe another review might be better suited. Ref added

Line 143 – failed instead of fail. Corrected

Line 254 – the ref.no.17 seems to be wrong. No, ref 17 is correct and is related  to ROS. 

References – Several numbers are missing (e.g. 97) and others are still with et al. after 3 authors

Ref 97 was not missing ... Giudice et al (Lancet). We paid attention to the other remark.

In conclusion, we guess that there was a fruitfull discussion between the reviewer and authors and we hope that, finally, it improved the ms.

Reviewer 2 Report (New Reviewer)

The review entitled “Emerging drug targets for endometriosis”, by Marie-Madeleine Dolmans and Jacques Donnez is of remarkable interest. The review is very well written and documented, in line of previous work of the authors. I recommend its publication in Biomolecules.

The figure is well conceived, but I suggest adding examples of each “targeting” type of molecule. I also suggest to include a short section about the current studies about pain relief treatments for endometriosis.

Author Response

Answers to referee 2:   Thank you for your efforts to revise the manuscript.

  • The figure is well conceived, but I suggest adding examples of each “targeting” type of molecule. Done. But, after having redrawn the figure with the Biorender program, we believe the figure is indeed interesting but very 'heavy', as there is a lot of information on it now (information that is in the text anyway). So we would like to suggest to keep the original one, as it goes, in our opinion, more 'to the point'. For your evaluation, we added the modified figure as suggested.
  • I also suggest to include a short section about the current studies about pain relief treatments for endometriosis. As we were asked to focus on potential  emerginig drug targets for endometriosis in ths review, it is true we did not write about pain treatments. Moreover, a complete review was recently dedicated to this symptom treatment, (see below the first objective of this article) and we did not want to repeat and do plagiarism. So we believe that readers interested in that topic can be referred to this article in JCM 2021.
  • --> Review J Clin Med,  2021;10(5):1085. doi: 10.3390/jcm10051085. Endometriosis and Medical Therapy: From Progestogens to Progesterone Resistance to GnRH Antagonists: A Review Jacques Donnez, Marie-Madeleine Dolmans  Abstract Background: The first objective of this review was to present, based on recent literature, the most frequently applied medical options (oral contraceptive pills (OCPs) and progestogens) for the management of symptomatic endometriosis, and evaluate their effectiveness in treating premenopausal women with endometriosis-associated pelvic pain, dysmenorrhea, non-menstrual pelvic pain and dyspareunia.

Reviewer 3 Report (New Reviewer)

This review is entitled "Emerging drug targets for endometriosis" an important topic for a disease with suboptimal treatments. Generally, the review appears to be comprehensive and well-referenced, and a timely contribution to the field summarizing this topic. However, I find that the writing and structuring of some sections could be improved. I have annotated some suggestions and queries in the attached file.

Author Response

Answer to comments of Reviewer 2:

Comment 1: GnRH antagonist was added.

Comment 2: We agree, corrected.

Comment 3: English of this section has been improved.

Comment 4: we agree. The words ‘stromal cells’ were added

Comment 5: ‘signal’ was added

Comment 6: we started a new paragraph.

Comment 7: section addressing inflammation: the concluding sentence has been modified.

Comment 8: Sections on inflammation have been put together, by renaming the previous sections, as suggested by the referee. Following sections have been renumbered accordingly.

Comment 9: It make sense, as the diminished activity of NK cells is responsible for deficiency in clearing endometriotic cells.

Comment 10: ‘several’ has been corrected.

Comment 11: non-coding miRNA

Comment 12: the sentence was completed

Comment 13: we agree. We modified accordingly.

Comment 14: To avoid confusion, this sentence was suppressed. The remaining sentences explain enough why these drugs were considered and suitable.

Comment 15: corrected.

Comment 16, p7: are expressed: corrected

Comment 17: we agree, and we have added that Elagolix has been approved by the FDA for the management of endometriosis.

Comment 18: editing needed: done.

Round 2

Reviewer 1 Report (Previous Reviewer 3)

The authors have addressed nearly all of my concerns properly. I only have some minor remarks:

Lines 27-29 I would like to suggest to rephrase the sentence like: Each of its three distinct pelvic forms ... presentations [2], but ... manifestations [3]. Because ref no 2 does only describe the three endometriotic entities.

Introduction first paragraph - Ref no 6 is missing

Line 35 please replace menopausal by menopausal-like

Line 393 please replace ...or in vitro... by ...or with in vitro...

Line 421 Please correct the typo [625] to [62]

Line 447 Please replace ...in other experiments...by ...in experiments...

Line 490 Please replace Noe by Noë

Lines 534-536 Please rephrase the sentence like: Several studies ... in the invasion process [92] and ... lesions [80,81,93]. Because as clearly stated in my last review ref no. 92 does only present collective cell migration but not nerve fibers as a cause of endometriosis.

Author Response

We thank the reviewer for his quick and thorough revision.

Our answers are in italics below.

-Lines 27-29 I would like to suggest to rephrase the sentence like: Each of its three distinct pelvic forms ... presentations [2], but ... manifestations [3]. Because ref no 2 does only describe the three endometriotic entities. DONE

-Introduction first paragraph - Ref no 6 is missing . CORRECTED

-Line 35 please replace menopausal by menopausal-like. As GnRH Ag/At induce hypo-oestrogenism, we prefer to use the word menopausal in this case.

-Line 393 please replace ...or in vitro... by ...or with in vitro... DONE

-Line 421 Please correct the typo [625] to [62]. DONE

-Line 447 Please replace ...in other experiments...by ...in experiments... DONE

-Line 490 Please replace Noe by Noë. CORRECTED

-Lines 534-536 Please rephrase the sentence like: Several studies ... in the invasion process [92] and ... lesions [80,81,93]. Because as clearly stated in my last review ref no. 92 does only present collective cell migration but not nerve fibers as a cause of endometriosis. DONE

This manuscript is a resubmission of an earlier submission. The following is a list of the peer review reports and author responses from that submission.

Round 1

Reviewer 1 Report

The paper presents a valuable review, focusing on new therapeutics for endometriosis.

As the authors mentioned, a few studies suggest the possibility of a new therapeutic drug. How did the authors select specific drugs included in this review among these scientific works? Any criteria? 

Reviewer 2 Report

In this manuscript, the authors tried to focus on the inflammation and inflammatory molecules, and discuss the perspectives of non-hormonal treatment of endometriosis, especially targeting inflammation, NF-Κb, cytokines, ROS, apoptotic and autophagic pathways, EMT, angiogenesis and neuroangiogenesis. This review is very interesting. However, there are major defects which need further modification.

Comments:

1.       Line 86-94, the title of 1.3.1 is inflammation and inflammatory molecules, however, the high levels of inflammatory molecules was ignored. Please add the information of common inflammatory molecules in endometriosis in this paragraph, and provide a table to describe it.

2.       Line 112-122, the possible reasons for different results of COX-2 inhibitors or TNF-a inhibitors in different reports should be discussed. And the effect of rhTNFR and anti-TNF monoclonal antibody in the treatment of endometriosis should be supplemented.

3.       The other roles of COX-2 in endometriosis should be introduced, such as endometriotic immune cell regulation.

4.       In the section of 2.2., the roles of NF-kB in the regulation of endometriotic stromal cells should be provided.

5.       Please add the figures and tables to introduce the targeted treatment in detail.

6.       The limitations of this study should be discussed.

Reviewer 3 Report

In the submitted review Dolmans & Donnez summarize emerging drug targets for endometriosis. Although the authors presented an exhaustive manuscript, several issues need to be addressed in more details.

Major issues

The authors did not provide any description of the methods they used, e.g. which keywords were used to search PubMed, how many papers were obtained and how did they evaluate which to integrate in the review? Furthermore a PRISM flow chart would be helpful.

Chapter 1.1. Progesterone resistance – Several important refs are missing as well as some new findings are not included. Flores et al. 2018 in JCEM 103:4561 showed that the progesterone status predicts the response to progestin therapy. This was discussed in two letter 2019;104:2147 and 2149. Furthermore, 2 reviews are not mentioned which are discussing in depth progesterone resistance (Trends Endocrinol Metab 2018; 29:535 and Hum Reprod Update 2020;26:565)

Chapter 1.1. line 67/68. The epigenetic theory is relatively new and should be discussed more critically, also the family predisposition and genetic causes have up to date not resulted in identification of any major candidate gene. Thus, there are some doubts whether it might not be simply the different genetic background which can also explain the family predisposition as described for rat/mice strains and tumors.

Chapter 1.2. The authors suggested that the differences between eutopic and ectopic endometrium might be due to different patterns of steroid receptors and/or progesterone resistance. This is a very simplistic view and there might be alternative hypotheses like e.g. the endometriotic disease theory by Koninckx et al. (Gynecol. Obstet. Investig. 1999, 47, 3). Furthermore, the differences between eutopic and ectopic endometrium are over-estimated (J Clin Med 9:E1915). Also it is not very fair to present a self-citation of the authors (ref. 34), which was only a letter to the editor (without a real good explanation why progesterone resistance might explain the heterogeneity), but the answer of the authors (Colgrave et al. 2021; Hum Reprod 36:2624) was not given.

Chapter 1.3. There are no refs to the crucial role and expression of COX1/2 in the endometrium and endometriosis. Furthermore, as recently shown by Morris et al. 2021 (Front Physiol 12:805784; Burns et al. 2012, 2018 Endocrinology 153:3960 and 159:103) with animal models of endometriosis, initiation of endometriosis is immune-dependent (neutrophils and macrophages) and only progression is hormone-dependent.   

Chapter 2.1. There are no arguments/observations given why COX-2 should be a target for therapy.

Chapter 2.5. No EMT regulators as possible therapy targets are mentioned. A recent review about EMT (J Clin Med 9:E1915) clearly summarized that there is only a partial EMT but no more which was further substantiated by Suda et al. 2019 in Hum. Reprod., 34, 1899.

Chapter 2.6. lines 238-240. Although the authors relied heavily on their own manuscripts (refs 77-81) I really wonder whether any of them could show “that nerve fibers participate considerably in the development and invasion process of endometriotic lesions.” First, only in ref. 78 something about the role of nerve fibers in the development of lesions could be found the other refs are mainly dealing with collective cell migration. However, the conclusion of ref. 78 is based only on immunohistochemistry but no causal relationship between nerve fibers and development of endometriotic lesions was presented. Thus, this paragraph need to be presented more cautiously.

Chapter 2.7. The main problem of the endometriosis patients and therapy failure seems to be the progesterone resistance due to impairment of the receptor or imbalance between PR-A and PR-B (Flores et al. 2018 JCEM 103:4561). Thus, why not thinking about how to overcome the PR impairment by triggering expression of the PR or something similar?

Chapter Conclusion – Again the epigenetic and genetic causes are not the smoking gun of endometriosis. Also EMT and transformation of endometrial cells into more motile cells; is this really necessary? Tissue breakdown is monthly, freeing the endometrial cells without the need to transformation like tumor cells. Furthermore, migration is not the main point, it is primarily contractions of the uterus and fallopian tubes which transport the cells to the pelvis. Last but not least I really wonder why the authors have nowhere discussed the advantages of local therapy versus oral applications.

Minor issues

Intro line 32 – Although many people believe that the prevalence is around 5-10% a recent review found that only 0.7-8.6% of women in the general population suffer from endometriosis J. Minim. Invasive Gynecol. 2020, 27, 452-461.

Intro line 32/33 – The 3 forms you mentioned are only in the pelvis, however, there are also extra-pelvic manifestations of the disease.

References – Refs 11, 26, 33, and 34 the months should be deleted. Ref. 48 please replace speciesinduced by species induced. Ref. 83 Please replace Obtset by Obstet. Refs 97, 98, 103, the et al. is already given after 3 authors whereas in the other refs it was done after 6 authors.